# A New Seven-Segment Profile Algorithm for an Open Source Architecture in a Hybrid Electronic Platform

José R. García-Martínez [†] , Juvenal Rodríguez-Reséndiz * and Edson E. Cruz-Miguel [†]

División de Investigación y Posgrado, Facultad de Ingeniería, Universidad Autónoma de Querétaro (UAQ), Cerro de las Campanas, S/N, Col. Las Campanas, Querétaro C.P. 76010, Mexico; jose.gm@uaq.mx (J.R.G.-M.); ecruz30@alumnos.uaq.mx (E.E.C.-M.)

\* Correspondence: juvenal@uaq.edu.mx; Tel.: +52-442-192-1200

† These authors contributed equally to this work.

**Abstract:** The velocity profiles are used in the design of trajectories in motion control systems. It is necessary to design smoother movements to avoid high stress in the motor. In this paper, the rate of change in acceleration value is used to develop an S-curve velocity profile which presents an acceleration and deceleration stage smoother than the trapezoidal velocity profile reducing the error at the end of the duty-cycle pre-established in one degree of freedom (DoF) application. Furthermore, a new methodology is developed to generate a seven-segment profile that works with negative velocity and displacement constraints applying an open source architecture in a hybrid electronic platform compounded by a system on a chip (SoC) Raspberry Pi 3 and a field programmable gate array (FPGA). The performance of the motion controller is measured through the comparison of the error obtained in real-time application with a trapezoidal velocity profile. As a result, a low-cost platform and an open architecture system are achieved.

**Keywords:** low-cost platform; FPGA; S-curve; motion control; robotics; SoC

## 1. Introduction

The velocity profiles have been studied broadly in recent years to design point-to-point trajectories in robot manipulators, conveyor belts, computer numerical control (CNC) machinery or whatever system with the use of direct current (DC) and alternating current (AC) motors [1,2]. Velocity profiles have an essential role in motion control since it is possible to accomplish a target position reducing the vibrations and the energy consumption, increasing the precision and the durability of the systems [3,4]. Nowadays, a great variety of velocity profiles exist, but their accuracy depends on the velocity's demeanor. Since, if the velocity changes abruptly, the behavior of acceleration could cause discontinuities in the trajectory [5]. The rate of change in acceleration is denominated as jerk [6], whether acceleration changes too fast, the vibrations increase their frequency causing damage to the structure of the mechanical system [7]. Therefore, when something like this happens, it is possible to deduce that the jerk value is too big, which means that the energy consumption is high.

There are different velocity profiles applied to specific processes, the common ones are the triangular velocity profile, parabolic velocity profile and the trapezoidal velocity profile [8]. The triangular velocity profile is a piece-wise defined function given by two linear segments corresponding to the acceleration and deceleration of the actuator, the jerk value is high because of the acceleration changes radically. On the other hand, the parabolic velocity profile presents a smoother velocity curve than the triangular, making the jerk value less than triangular profile but neither of them maintain a velocity constant phase [9,10], it means that they accelerate and decelerate the actuator immediately. The trapezoidal velocity profile consists of three phases: acceleration,

constant velocity and deceleration phase [10,11]. A constant velocity phase offers less wear on the actuator extending its life period, since, the change in acceleration occurs after a period and not abruptly after to reach the desired velocity [12,13]. The trapezoidal velocity profile is the most used in the industry, although the jerk value is high-rise [5,14]. A second-degree polynomial describes the transition of the position making accessible the implementation in an embedded system due to the low level of processing [15,16]. On the other hand, the seven-segment velocity profile, known as the S-curve velocity profile, has been studied broadly in recent years obtaining better results than other profiles because of the jerk value takes a constant amount [17], decreasing the damage produced by high-frequency vibrations in the structure of the system. The implementation in an embedded system can need a complex architecture to support the seven position's equations defined by a third-degree polynomial.

The controller is fundamental to the application of a motion control system [18,19], the velocity profile contributes with several points that describe the path, but the controller must follow, reducing the error significantly, the trajectory [20,21]. According to a comparison presented in [2], the controller and the profile generator algorithm can be chosen by the designer according to its experience. For instance, Jeong et al. [22] proposed an algorithm that can determine the coefficients of jerk limited profiles, but it only works with non-negative velocity and displacement constraints. Wang Bangji et al. [23] developed a velocity profile algorithm for stepper motor controller in an field programmable gate array (FPGA) where the characteristics of the stepper motor are introduced by the user to generate the velocity profile. The algorithm is restricted to other motors. Tou Wai Kei et al. [24] designed a speed regulator by implementing an S-curve velocity profile in a microcontroller to control an elevator featuring direct landing. The profile generator only accepts the maximum acceleration, an initial jerk value, and a maximum velocity. Working with microcontrollers can presents problems at the moment to migrate the algorithm to a different family of microcontrollers due to the internal architecture used among them, in the FPGA the code is preserved, it does not change and it can be migrated in an easy way without altering the algorithm. This paper presents a new methodology to obtain the S-curve coefficients in real-time, which includes the non-negative velocity and displacements constraints applied to DC motors using an open architecture based on a Raspberry Pi 3 combined with an FPGA which is a low-cost platform compared to closed architectures on the market capable to generate trajectories. Furthermore, the user can introduce the total displacement, the duration of the movement and the length of the acceleration-deceleration phases. Besides, the profile generator calculates the acceleration and jerk parameters according to the desired position.

## 2. Background

### 2.1. General Model of the Seven-Segment Velocity Profile

The motion planning designer must develop smoother trajectories to avoid discontinuities in the acceleration and reduce the strain and exertion on the actuators and the mechanical architecture [25,26]. Since a third-degree polynomial models the behavior of the position, the seven-segments velocity profile offers the possibility to maintain the jerk with a constant value, obtaining a step profile for the rate of change in acceleration [27]. An important aspect to mention is that, when the degree of the polynomial is increased or decreased, the profiles tend to shift themselves into a particular position, adopting a different shape. For instance, in Figure 1c, the acceleration has taken the shape of a trapezoidal profile allowing linear changes at the acceleration, even the velocity adopts a continuous shape connected by parabolic-blends in the form of S-curve for the acceleration phase and an inverse S-curve for the deceleration stage.

It is necessary to analyze Figure 1 to compute the values of the $a_0$, $a_1$, $a_2$ and $a_3$ parameters of Equation (1), by seven-segments with an interpolation method [1]; each segment represents an equation depending on the motion profile under analysis. So, it is acceptable to propose a constant

value for the jerk according to a given acceleration, but it must exist a relationship with the maximum amount of the speed given by the data-sheet of the actuator.

$$\theta_d(t) = a_0 + a_1 t + a_2 t^2 + a_3 t^3, \tag{1}$$

where $\theta_d$ is the aim position. When the maximum values of the jerk, acceleration, and velocity are known, the Equations (3)–(5) can be applied to generate the wished trajectory with the seven-segments velocity profile. The total duration of the movement $T$ is also known, and it is provided by the designer. As mentioned before, the jerk has a step profile because it retains its value as a constant, so that in Figure 1d one can see how it varies with respect to the time and get (2).

$$J(t) = \begin{cases} j_{max} & t \in [0, T_{s1}) \\ 0 & t \in [T_{s1}, T_{s2}) \\ j_{min} & t \in [T_{s2}, T_{s3}) \\ 0 & t \in [T_{s3}, T_{s4}) \\ j_{max} & t \in [T_{s4}, T_{s5}) \\ 0 & t \in [T_{j5}, T_{s6}) \\ j_{min} & t \in [T_{s6}, T]. \end{cases} \tag{2}$$

Assuming that $j_{min} = -j_{max}$ in four periods, $T_{si}$ is the length of the $i$-th segment, where $i = 0, 1, 2, ..., 6$. The acceleration is calculated after integrating (2), segment by segment. Equation (3) is the general equation to get acceleration. Equation (4) is the equation of the velocity and (5) is the equation to obtain the position.

$$\alpha(t) = \alpha(T_{si}) + \int_{T_{si}}^{T} J(\tau_i) d\tau_i \tag{3}$$

$$\omega(t) = \omega(T_{si}) + \int_{T_{si}}^{T} \alpha(\tau_i) d\tau_i \tag{4}$$

$$\theta(t) = \theta(T_{si}) + \int_{T_{si}}^{T} \omega(\tau_i) d\tau_i. \tag{5}$$

The relative time parameter of the integral is defined as $\tau_i = T - T_{si}$ where $i = 1, 2, 3, ..., 6$ represents the segments of the displacement. The result of the integration of (3) is the acceleration profile showed in Figure 1c. The acceleration shows a linear variation until reach a constant value, and then presents a linear deceleration.

In order to draw the acceleration profile with the desired characteristics, it is necessary to substitute the $a_{acc}$ and the $j_{max}$ values in (6), which is the result of the integration of (3). Notice that the acceleration phase goes from the origin to $T_{s3}$ in Figure 1c, whereas the deceleration phase $(T - T_{s4})$ is compound by an inverse trapezoid.

$$\alpha(t) = \begin{cases} j_{max} T_{s1} & t \in [0, T_{s1}) \\ a_{acc} & t \in [T_{s1}, T_{s2}) \\ a_{acc} + j_{min}(T_{s3} - T_{s2}) & t \in [T_{s2}, T_{s3}) \\ 0 & t \in [T_{s3}, T_{s4}) \\ j_{max}(T_{s5} - T_{s4}) & t \in [T_{s4}, T_{s5}) \\ a_{dec} & t \in [T_{s5}, T_{s6}) \\ a_{dec} + j_{min}(T - T_{s6}) & t \in [T_{s6}, T] \end{cases} \tag{6}$$

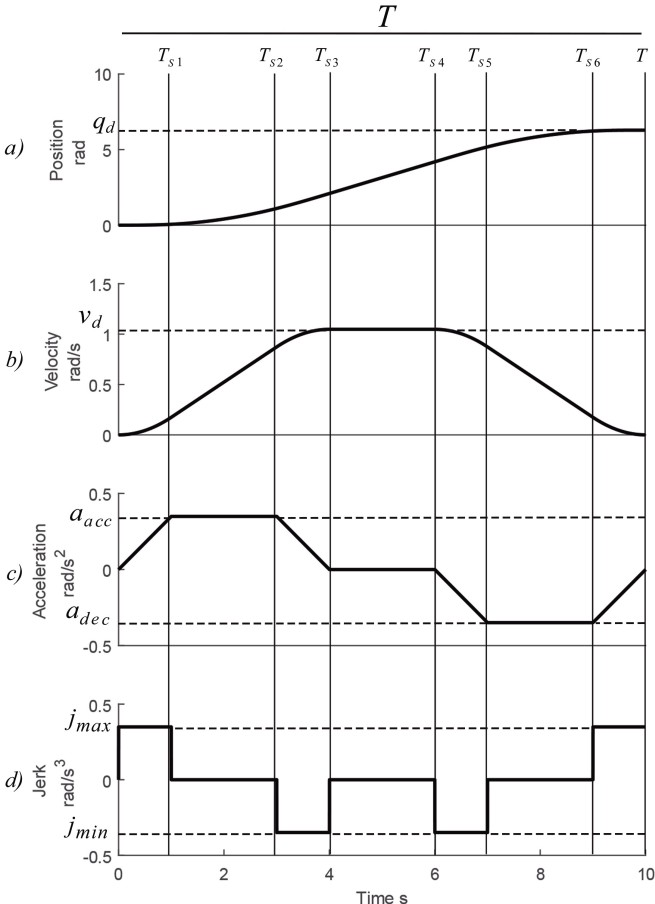

**Figure 1.** Motion profiles by given a set-point, (**a**) position, (**b**) velocity, (**c**) acceleration and (**d**) jerk.

According to (4), it is possible to compute the velocity profile of each segment from the integral of the acceleration. The seven-segments velocity profile is modeled by (7).

$$
\omega(t) = \begin{cases}
v_1 + \frac{j_{max}}{2}(\tau_1)^2 & t \in [0, T_{s1}) \\
v_1 + \frac{j_{max}}{2}T_1^2 + a_{acc}\tau_2 & t \in [T_{s1}, T_{s2}) \\
v_1 + \frac{j_{max}}{2}T_1^2 + a_{acc}T_2 + a_{acc}\tau_3 + \frac{j_{min}}{2}\tau_3^2 & t \in [T_{s2}, T_{s3}) \\
v_{max} & t \in [T_{s3}, T_{s4}) \\
v_{max} - \frac{j_{max}}{2}(\tau_5)^2 & t \in [T_{s4}, T_{s5}) \\
v_{max} - \frac{j_{max}}{2}T_5^2 - a_{dec}\tau_6 & t \in [T_{s5}, T_{s6}) \\
v_{max} + \frac{j_{max}}{2}T_5^2 - a_{dec}T_6 + a_{dec}\tau_7 + \frac{j_{min}}{2}(\tau_7)^2 & t \in [T_{s6}, T].
\end{cases}
\tag{7}
$$

Here, $\tau_{i-1} = T - T_{si}$ is the relative time of each segment of (7), $i = 1, 2, ...7$ defines the segment under analysis and $T_i$ is the duration of the *i*-th stage. The acceleration, when the maximum speed $v_{max}$ is constant, turns to zero. Finally, it is necessary to solve (5), to compute the position profile for the *i*-th segment according to Figure 1a. Equation (8) represents the piece-wise behavior of the position respect the time. Notice that it posses a third-degree polynomial equations that modeled the desired position in certain segments.

$$\theta(t) = \begin{cases} q_1 + v_1\tau_1 + \frac{j_{max}}{6}(\tau_1)^3 & t \in [0, T_{s1}) \\ q_2 + v_2\tau_2 + \frac{a_{acc}}{2}(\tau_2)^2 & t \in [T_{s1}, T_{s2}) \\ q_3 + v_3\tau_3 + \frac{a_{acc}}{2}(\tau_3)^2 - \frac{j_{min}}{6}(\tau_3)^3 & t \in [T_{s2}, T_{s3}) \\ q_4 + v_4\tau_4 & t \in [T_{s3}, T_{s4}) \\ q_5 + v_5\tau_5 - \frac{j_{max}}{6}(\tau_5)^3 & t \in [T_{s4}, T_{s5}) \\ q_6 + v_6\tau_6 - \frac{a_{dec}}{2}(\tau_6)^2 & t \in [T_{s5}, T_{s6}) \\ q_7 + v_7\tau_7 + \frac{a_{dec}}{2}(\tau_7)^2 - \frac{j_{min}}{6}(\tau_7)^3 & t \in [T_{s6}, T], \end{cases} \tag{8}$$

where

$$\begin{cases} q_2 = q_1 + v_1 T_1 + \frac{j_{max}}{6}T_1^3 \\ q_3 = q_2 + v_2 T_2 + \frac{a_{acc}}{2}T_2 \\ q_4 = q_3 + v_3 T_3 + \frac{a_{acc}}{2}T_3^2 - \frac{j_{min}}{6}T_3^3 \\ q_5 = q_5 + v_5 T_5 - \frac{j_{max}}{6}T_5^3 \\ q_6 = q_5 + v_5 T_5 - \frac{j_{max}}{6}T_5^3 \\ q_7 = q_6 + v_6 T_6 - \frac{a_{dec}}{2}T_6^2, \end{cases}$$

when the motor has reached the maximum velocity $v_{max}$ using the desired position $\theta_d$ and duration of the movement $T$ proposed, it maintains a constant speed and the position in that phase is a third-degree polynomial.

## 2.2. Proposed Method to Compute the Desired Jerk

It was necessary to get access to the data-sheet of the actuator to design a point-to-point trajectory [1], and check for some specific characteristics as torque, minimum and maximum values for the velocity and acceleration, and so on [28,29]. These parameters can be used by the designer for solving a determined task using delimiter parameters to prevent damage caused under dynamic loads initiated by molecular bond separation in the material, and to reduce the vibrations on the actuator [25,30,31]. The aim of working with a range of velocities was to assign the desired speed and compute the needed values of the acceleration and the jerk respect to the rate of change in position parameter proposed. A condition to satisfy is presented as follow.

$$\omega_d = \omega(T_{si}) + \int_{T_{si}}^{T} \alpha(\tau_i)d\tau_i \leq \omega_{max}, \tag{9}$$

where $\omega_d$ and $\omega_{max}$ are scalar of the desired and the maximum speed permitted by the DC motor respectively and they must satisfy (9). Whether a more significant value is declared for $\omega_d$ than $\omega_{max}$, the actuator is going to try to reach that speed, demanding a higher voltage than the provided by the manufacturer. Therefore, an approximation for the planning of the seven-segments velocity profile consists of defining the values for the rate of change of position and acceleration over a function based on the acceleration-deceleration stage proposed. The total time for acceleration phase is represented in (10).

$$T_{acc} = T_{s1} + T_{s2} + T_{s3} \tag{10}$$

The phase when the speed is varying respect the time is constituted by three segments, two parabolic-blends and a linear displacement related to the speed profile. Assuming that the acceleration phase is symmetric respect the deceleration phase, it is possible to suppose that $T_{acc} = T_{dec}$. In order to compute $T_{acc}$ it is necessary to multiply the total time of motion $T$ by a factor $\gamma \in R$, where $0 \leq \gamma \leq \frac{1}{2}$. So, the acceleration time can be calculated using $T_a = \gamma T$. A value of acceleration and jerk have to be calculated to use Equations (6)–(8),. The velocity obtained by the desired position can be computed using (11).

$$\omega_d = \omega(T_{s3}) + \int_{T_{s3}}^{T_{s4}} \alpha(\tau_3)d\tau_3 = \frac{\theta_d}{(1-\gamma)T}. \tag{11}$$

The result of (12) took the value of the $\omega_d$ as the maximum velocity computed with the desired position $\theta_d$. Therefore, $\omega_d = \omega_{max}$ in the segment $\{T_{s3}, T_{s4}\}$ over Figure 1b. The parameter $\theta_d$ is an scalar and its range is $(R^-, R^+)$. The segment of time when the jerk kept a constant amount $T_{jerk}$ is less than the acceleration phase $T_{acc}$, it means $T_{acc} \geq T_{jerk}$. It must exist a relationship between the $T_{acc}$ and $T_{jerk}$ to ensure the continuity of Equations (6)–(8), so that, the length of the acceleration phase has to be multiplied by a factor $\varphi \in R$. $\varphi$ can take values in the range $0 \leq \varphi \leq \frac{1}{2}$, thus, the constant value of the jerk should endure $T_{jerk} = \varphi T_{acc}$. Supposing that the acceleration phase is symmetric to the deceleration phase, they have the same duration $T_{acc} = T_{dec}$ with an opposite magnitude, solving (12) to determine the acceleration value.

$$\alpha_d = \alpha(T_{s1}) + \int_{T_{s1}}^{T_{s2}} J(\tau_1)d\tau_1 = \alpha(T_{s5}) + \int_{T_{s5}}^{T_{s6}} J(\tau_5)d\tau_5 = \frac{\theta_d}{\gamma(1-\gamma)(1-\varphi)T^2}. \tag{12}$$

where $\alpha_d$ is the maximum value reached with the desired position parameter in total duration motion proposed. The $\alpha_d$ is a constant parameter, $\alpha_d \in R$. The jerk value can be computed by (13).

$$J_d = \frac{\alpha_d}{T_{jerk}} = \frac{\theta_d}{T_{jerk}\gamma(1-\gamma)(1-\varphi)T^2}. \tag{13}$$

Here $J_d$ is the constant jerk value, $J_d \in R$. Notice that exists a dependence among the values of the velocity $\omega_d$, acceleration $\alpha_d$ and jerk $J_d$, respect to the total time of the motion an the target position. Once the jerk is obtained, it can be substituted in (2) to draw the jerk profile. The desired position, using (8), can be computed with Equation (14).

$$\theta_d = q_6 + v_6T_6 + v_7T - \frac{a_{dec}}{2}T_6^2 + \frac{a_{dec}}{2}(T)^2 - \frac{j_{min}}{6}(T)^3. \tag{14}$$

The value of $\theta_d$ is calculated from the last stage of (8) and represents the total displacement from the movement. It is necessary to calculate a negative acceleration and jerk parameters to compute the negative velocity constraints, using Equations (13) and (14). $J_{inv} = sgn(j_{si} - J_d) \mid J_d \mid$ indicates if the sign of the jerk profile is positive or negative. Whether the jerk value is negative, the jerk profile presented in Figure 1d changes its shape to Figure 2b. On the other hand, inverting the jerk profile, one can obtain a negative acceleration since the jerk is negative. So, the stage sign for the acceleration can be obtained using the following constraint $\alpha_{inv} = sgn(j_{si} - J_d) \mid \alpha_d \mid$, where a negative sign for the acceleration profile correspond to invert the trapezoidal shape of acceleration, see Figure 2a. Using the negative acceleration, the velocity profile presents an inverted S-curve, Figure 3b.

Equations (2)–(8) are inverted after computing the desired jerk and the maximum acceleration needed to reach the jerk condition. It is added a new variable denominated as the initial position $\theta_0$ to compensate the total displacement writing in (15). Once the jerk and acceleration are computed for a negative displacement, in accordance with the actual value, it is possible to use the negative value of the computed jerk to generate the inverse trajectory.

$$\theta_{d'} = \theta_0 + \theta_d, \tag{15}$$

where $\theta_{d'}$ is the final displacement if the initial position $\theta_0$ is the last movement reached for the shaft of the motor, it means that the new desired position has moved in an opposite way, and the total displacement computed from the home point is described in (16):

$$\theta_d = q_0 - q_6 - v_6T_6 - v_7T + \frac{a_{dec}}{2}T_6^2 - \frac{a_{dec}}{2}(T)^2 + \frac{j_{min}}{6}(T)^3 \tag{16}$$

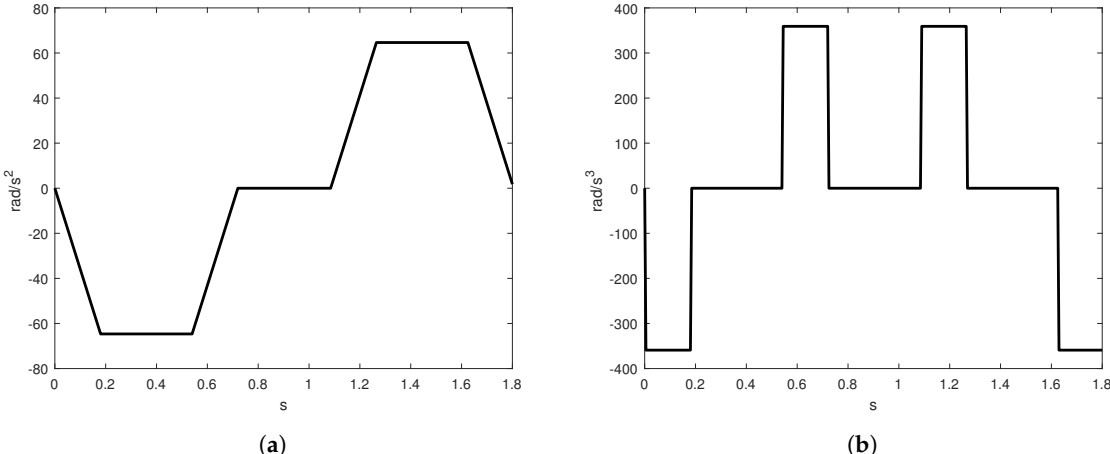

**Figure 2.** (**a**) Inverse trapezoidal acceleration profile, (**b**) inverse jerk profile.

The negative position is used to return the shaft of the motor to the initial position or to move it in an opposite way to the positive axis, the position profile is presented in Figure 3a.

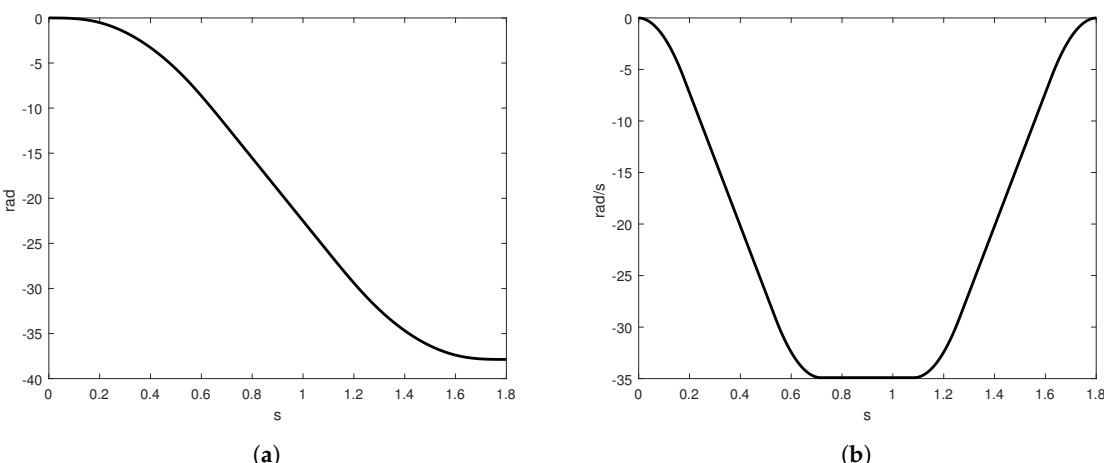

**Figure 3.** Inverse S-curve velocity profile (**a**) position, (**b**) S-curve velocity profile.

## 3. Methods and Experimentation

Motion control applications encompass an extensive range of topics related to the control of electromechanical systems, trajectories design to reduce the error increasing the precision of the system and handle of vibrations to minimize damage in mechanical structures. A new methodology is proposed in Figure 4 to reduce the error obtained experimentally applying a motion profile. In order to design a motion controller using a hybrid system compounded by the interaction of a Raspberry Pi 3 and an FPGA ZYBO-ZYNQ XC7Z010 from the family XILINX.

The idea of designing a motion controller with those architectures arises from the technological advances in industry. Using a single-board computer reduces the work-space and increases the possibility about everybody can interact with the system, making interactive the process to the operator in charge of the mechanical system. The processing and control systems are embedded on the Raspberry Pi 3. For the power stage, a pulse-width-modulation (PWM) servo drive model 12A8 from the family advanced motion Control is used [32].

The seven-segment velocity profile is developed in C programming; the user can set the target position $\theta_d$, the total time of displacement $T$, and the length of the acceleration-deceleration phases to compute the speed needed to reach the set-point and the jerk value.

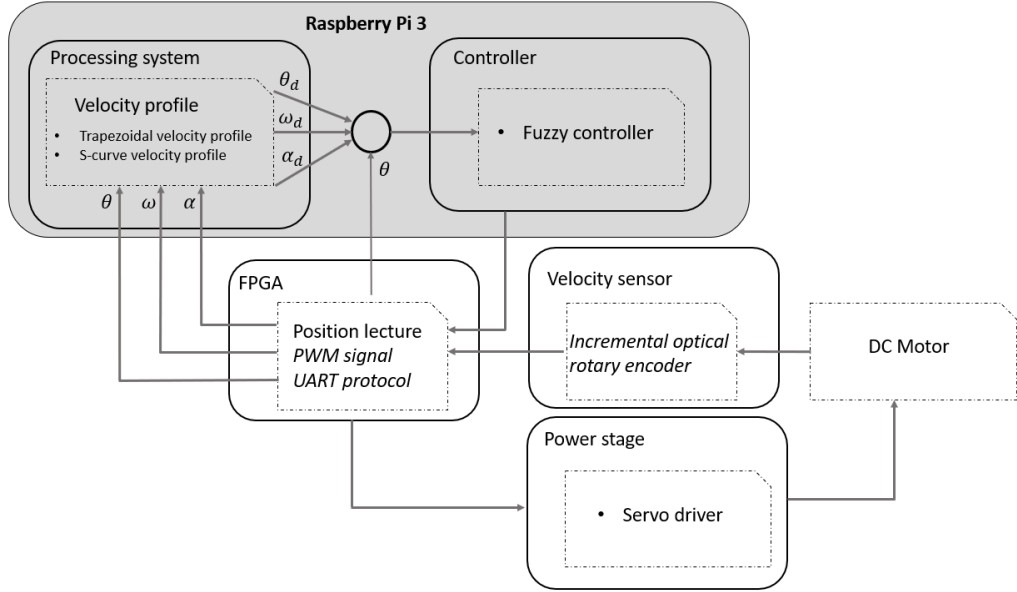

**Figure 4.** Hybrid electronic topology based on field programmable gate array (FPGA)-system on a chip (SoC).

On the other hand, the FPGA reads the real position of the shaft using a rotary encoder (2000 pulses per revolution). A quadrature signal was monitored each 20 na and it was stored in a register each 5 ms to send the data directly to the interface using the universal asynchronous receiver-transmitter (UART) protocol, Figure 5 shows the sequential logic implemented in the FPGA. $DATA\_Tx$ is the data to send, compounded by 8 bits; for this case, the encoder counts have a bandwidth of 16 bits. Rx and Tx are the communication lines, BaudRate (bits per second) is the transmission speed, it is important configure the same speed at 115,200 bauds in the C program of the Raspberry Pi 3. $DATA\_Rx$ is a buffer where the control signal is received, $eo\_Tx$ and $eo\_Rx$ are flags that indicate the end of transmission and reception of 1 byte, respectively.

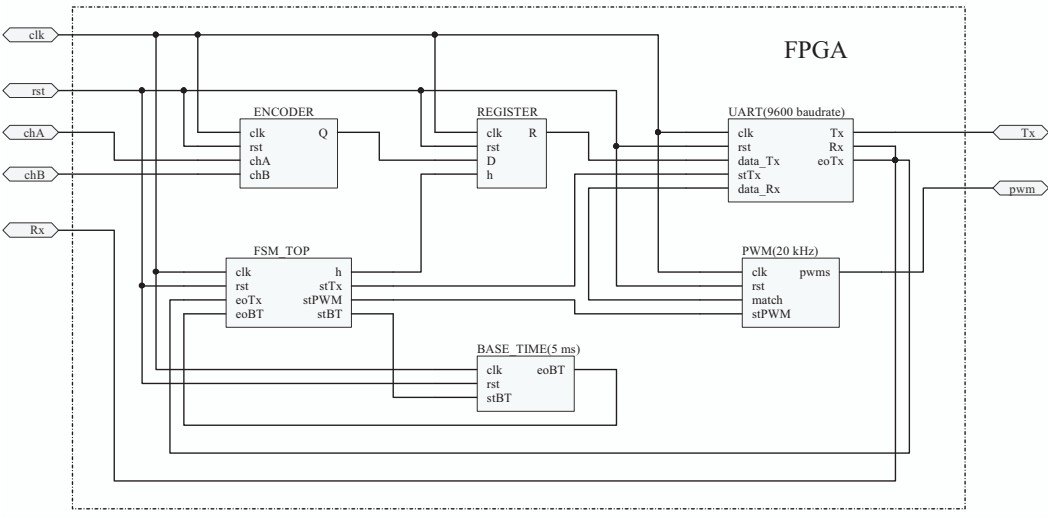

**Figure 5.** Entities embedded on the FPGA.

The Raspberry Pi 3 generated the trajectory parameters in real-time, using a sample time $T_s = 5$ ms, and sent them to the controller to minimize the error. The control signal was transmitted through the general purpose input–ouput (GPIO's) of the Raspberry Pi 3 to the FPGA in the form eight bits of information, the FPGA receives the data and transforms the control signal into PWM signal in order to send it to the servo drive to control the position of the motor.

*S-Curve Velocity Profile Parameters*

The S-curve velocity profile implementation was developed using (2)–(5). There were several considerations at the moment to define a trajectory such as: (a) total duration for the acceleration-deceleration must be equal for both stages $T_{acc} = T_{dec}$, (b) the magnitude of the acceleration $\alpha_d$ was obtained by the desired position $\theta_d$, (c) the jerk value is obtained with the $\alpha_d$ magnitude and $\theta_d$, all those values are calculated with the total duration of the movement $T$.

The parameters proposed for the design of the S-curve implementation are $\theta_d = 12\pi\ rad$ and $T = 1.8$ s, notice that the speed is calculated from the jerk value or using (9). Once the length of the movement was known, the acceleration-deceleration time was computed. For this application, the total time of the acceleration is divided by $\gamma = \frac{4}{10}$, and $T_{acc} = \gamma T$ in order to have a symmetric profile. On the other hand, the duration of the jerk stage must be divided by four times the acceleration phase $T_{acc}$ to ensure an S-curve velocity profile equal in length of acceleration, maximum velocity and deceleration stages, the proportional value was $\varphi = \frac{1}{4}$, so that the jerk was going to remain zero for $T_{jerk} = \frac{2}{4}T_{acc}$. The constant velocity stage had a duration of $\frac{2}{10}T$. The total distribution of the duration is computed in (17).

$$T = \underbrace{\frac{4}{10}T}_{T_{acc}} + \frac{2}{10}T + \underbrace{\frac{4}{10}T}_{T_{dec}}, \tag{17}$$

where

$$T_{acc} = T_{dec} = 0.72 \text{ s.}$$

The acceleration time was compounded by three phases of the jerk time, as presents (18). When $\frac{T_{acc}}{2}$ the jerk value turns to zero, it corresponds to the time interval $T_{s1} - T_{s2}$ from Figure 1d.

$$T_{acc} = \underbrace{\frac{T_{acc}}{4}}_{T_{jerk}} + \frac{T_{acc}}{2} + \underbrace{\frac{T_{acc}}{4}}_{T_{-jerk}}, \tag{18}$$

where

$$T_{jerk} = 0.18 \text{ s.}$$

The derivative respect the time of speed is calculated using (3). The parameter $\alpha_d$ depends on $\theta_d$, $T_{jerk}$, $T$, and the proportional constants of time $\gamma$ and $\varphi$. The magnitude of the acceleration is obtained from (12), so, using (19), $\alpha_d$ is computed.

$$\alpha_d = \frac{\theta_d}{\gamma(1-\gamma)(1-\varphi)T^2} = \frac{12\pi\ rad}{0.5832\ s^2} = 64.6418 \text{ rad/s}^2. \tag{19}$$

To estimate the value of the jerk, Equation (13) was used. Since the acceleration $\alpha_d$ has been obtained and the jerk interval $T_{jerk}$ is known, it was possible to obtain the rate of change in acceleration in (20).

$$J_d = \frac{\alpha_d}{T_{jerk}} = \frac{64.6418\ rad/s^2}{0.18\ s} = 359.1212 \text{ rad/s}^3. \tag{20}$$

In Table 1 is presented all the parameters needed to generate the algorithms in order to compute the trajectory proposed.

**Table 1.** S-curve parameters used for implementation.

| Parameters | | Values |
|---|---|---|
| Desired position | $\theta_d$ | $12\pi$ |
| Time of displacement | $T$ | 1.8 s |
| Time factor for acceleration time | $\gamma$ | 0.4 |
| Time factor for jerk phase | $\varphi$ | 0.25 |
| Acceleration time | $T_{acc}$ | 0.72 s |
| Deceleration time | $T_{dec}$ | 0.18 s |
| Velocity | $\omega_d$ | 34.9065 rad/s |
| Acceleration | $\alpha_d$ | 64.6418 rad/s$^2$ |
| Jerk | $J_d$ | 359.1212 rad/s$^3$ |

## 4. Simulation and Results

The algorithm proposed in this paper has been compared with the results presented in [33], where a fourth order polynomial S-curve is developed, the input data has been displayed in Table 2. For simulation, two lengths of acceleration phase were used. By one hand, a wide stage of acceleration was chosen with a factor $\gamma = \frac{2}{5}$; on the other hand, a short stage of acceleration with a factor $\gamma = \frac{1}{5}$ was used to compare the jerk response of both motion profiles. Besides, two target positions were chosen and taken from simulation section of [33], $\theta_{d1} = \frac{2\pi}{3}$ and $\theta_{d2} = \frac{\pi}{3}$.

**Table 2.** Simulation results compared from the [33] method.

| | | $\gamma = \frac{2}{5}$ | | $\gamma = \frac{1}{5}$ | | Yi Fang and Wenhai Liu [33] | |
|---|---|---|---|---|---|---|---|
| Position (rad) | Initial point | 0 | 0 | 0 | 0 | 0 | 0 |
| | Final point | $\frac{2\pi}{3}$ | $\frac{\pi}{6}$ | $\frac{2\pi}{3}$ | $\frac{\pi}{6}$ | $\frac{2\pi}{3}$ | $\frac{\pi}{6}$ |
| Kinematics constraints | Velocity (rad/s) | 2.319 | 0.5799 | 1.736 | 0.434 | 8 | 5 |
| | Acceleration (rad/s$^2$) | 5.137 | 1.2834 | 8.633 | 2.158 | 10 | 8 |
| | Jerk (rad/s$^3$) | 22.3 | 8.533 | 29.36 | 21.53 | 30 | 20 |

In Figure 6b, one can see the behavior of the velocity. For factor $\gamma = \frac{2}{5}$, the S-curve profile maintain symmetric intervals for acceleration-deceleration phases and maximum velocity. For $\gamma = \frac{1}{5}$ the acceleration-deceleration stages are shorter than the velocity phase. The position in Figure 6a adopts a different shape because of the acceleration-deceleration phase. Evaluating the response of the velocity profile for both factors, using an execution time of 1.5 s and a desired position of $\frac{2\pi}{3}$, the maximum speed for $\gamma = \frac{2}{5}$ is $\omega_{d1} = 2.319$ rad/s. On the other hand, the maximum velocity reached for the motion profile with a factor $\gamma = \frac{1}{5}$ is $\omega_{d2} = 1.736$ rad/s. The magnitude of the velocity obtained in [33] is bigger compared with the results obtained with the two factors proposed. The same occurred when the desired position was changed.

As mentioned before, factors $\gamma = \frac{1}{5}$ and $\gamma = \frac{2}{5}$ affect directly to the acceleration-deceleration phases. It means, when the length of the acceleration stage was short, the magnitude of the acceleration increased considerably. Figure 7a shows the behavior of the acceleration respect with each factor. Notice that, whether the acceleration-deceleration phases are short, the jerk tends to increase in magnitude as is displayed in Figure 7b. The jerk value was computed with respect to $\theta_d$ and $T$. The maximum jerk values were obtained for short acceleration-deceleration using $\gamma = \frac{1}{5}$ and $\theta_d = \frac{2\pi}{3}$. Unlike the jerk values obtained with the Yi Fang and Wenhai Liu algorithm $J_{max} = 30$ rad/s$^3$, the proposed algorithm performed a jerk value $J_{max} = 29.36$ rad/s$^3$. On the other hand, the acceleration stage in [33] is compounded by seven intervals, while the proposed method in this paper has three phases showing better simulation results. The length of the acceleration phase allows to reduce or increase the magnitude of the jerk, for a factor $\gamma < \frac{2}{5}$ the magnitude increases, while for a factor $\gamma = \frac{2}{5}$ the magnitude decreases, these factors can be chosen by the path planning designer.

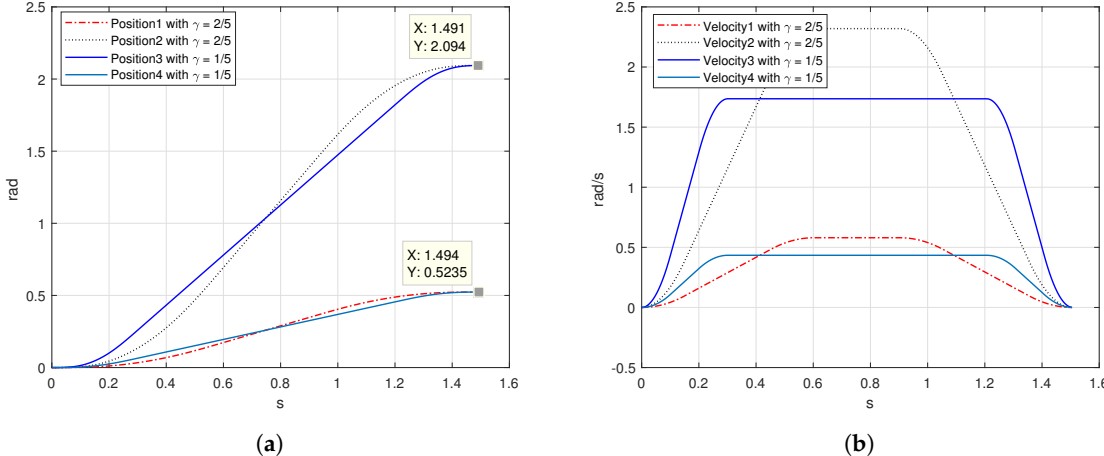

**Figure 6.** Position $\theta_d$ and S-curve velocity profile simulation with $\gamma = \frac{1}{5}$ and $\gamma = \frac{2}{5}$ factors (**a**) position, (**b**) velocity.

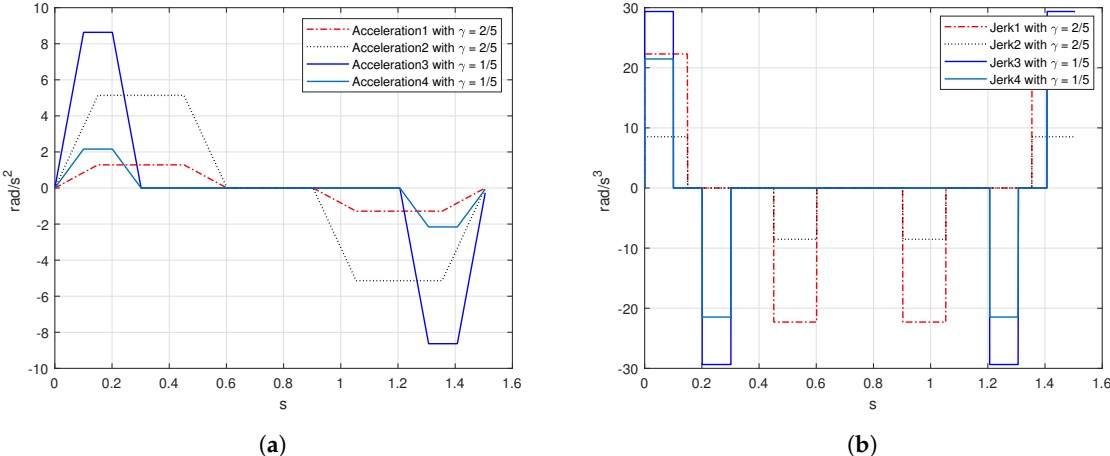

**Figure 7.** Acceleration and jerk simulation with $\gamma = \frac{1}{5}$ and $\gamma = \frac{2}{5}$ factors (**a**) acceleration, (**b**) jerk.

In this section, trapezoidal and S-curve velocity profiles were implemented to compare the response in real-time of the behavior of velocity and to measure the error in position of both motion profiles. A cylindrical load of 0.300 kg with inertia of 0.00011344 kg·m² was coupled to the shaft of a DC motor, as shows Figure 8, to obtain the experimental results. The motor has to compensate its movement even with the load to achieve the desired position $\theta_d$ following the trajectory computed by the motion profile.

### 4.1. Trapezoidal Velocity Profile

The trapezoidal velocity profile is the most used in industrial applications due to the ease of implementation since it consists of two linear equations describing the acceleration-deceleration phases, and a constant velocity stage [34]. The change in velocity was radical, so, the change in acceleration tended to infinity, mathematically speaking, but in real-world applications, the jerk permits an increment in residual vibrations, and damage in motors in certain period.

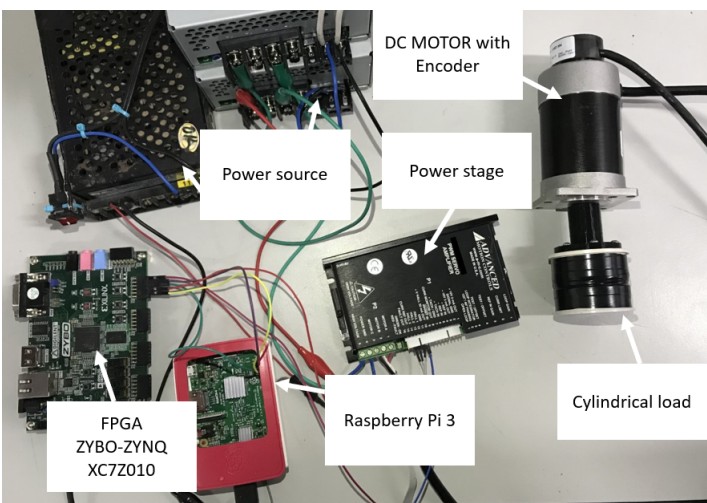

**Figure 8.** Low-cost platform based on FPGA-SoC.

The trapezoidal velocity profile presented in this section must reach a desired position $\theta_d = 12\pi$ rad in a total period of $T = 1.8$ s with a maximum velocity $\omega_{d_T} = 32$ rad/s. The trapezoidal velocity profile experimentally obtained is presented in Figure 9b.

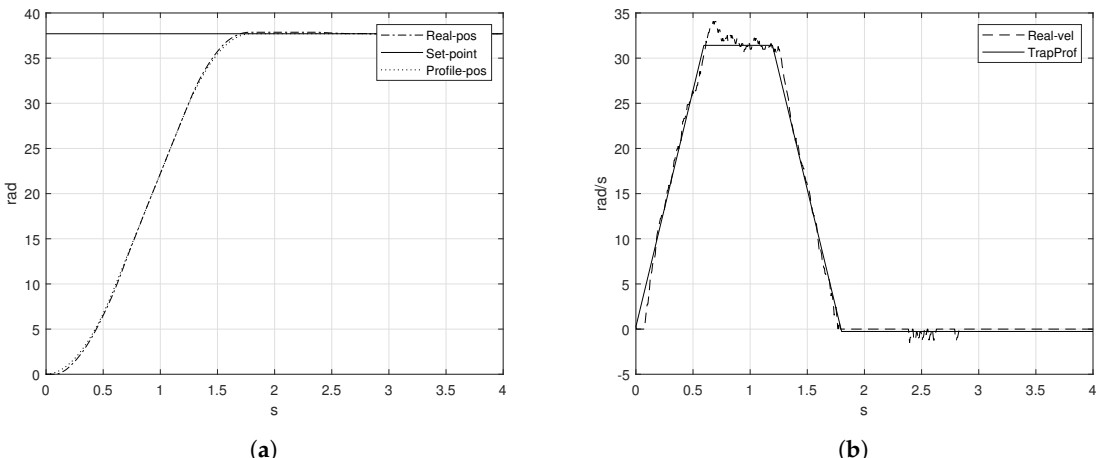

(**a**)                                                                 (**b**)

**Figure 9.** Trapezoidal velocity profile implementation (**a**) position, (**b**) velocity.

As the Figure 9b shows, the velocity follows the shape of the desired trapezoidal velocity profile with a disturbance when the speed has to be constant. The disturbance increased since the speed changed all of the sudden, so that the motor can not react instantly to follow the rate of change in position. The maximum peak of the real velocity goes to 34.3 rad/s although the desired position was reached, Figure 9a, there exists an error of $e_T = 0.18$ rad when the position should have reached the set-point in $T$. The error signal of the trapezoidal velocity is shown in Figure 10a. On the other hand, the maximum voltage required to achieve the maximum speed is about $u(t) = 1.39$ V and the control signal is presented in Figure 10b.

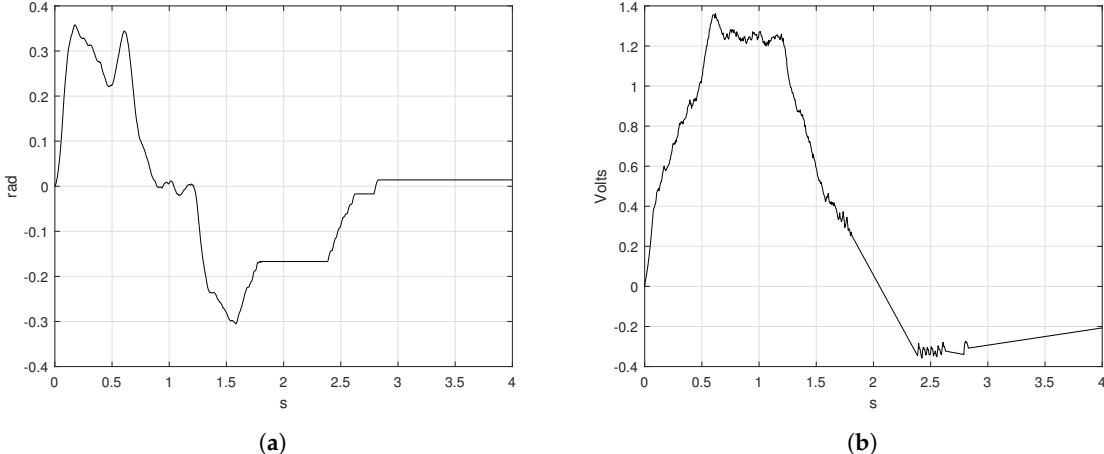

(a)                                                                    (b)

**Figure 10.** Error and control signals obtained from the trapezoidal velocity profile implementation (**a**) error position and (**b**) control signal.

## 4.2. S-Curve Velocity Profile Implementation

The implementation of the S-curve velocity profile was applied in the low-cost platform using the values presented in Table 1 and the methodology proposed in Section 3. Notice that the parameters proposed for the S-curve profile were similar to the trapezoidal velocity profile. The S-curve velocity profile obtained experimentally is presented in Figure 11b.

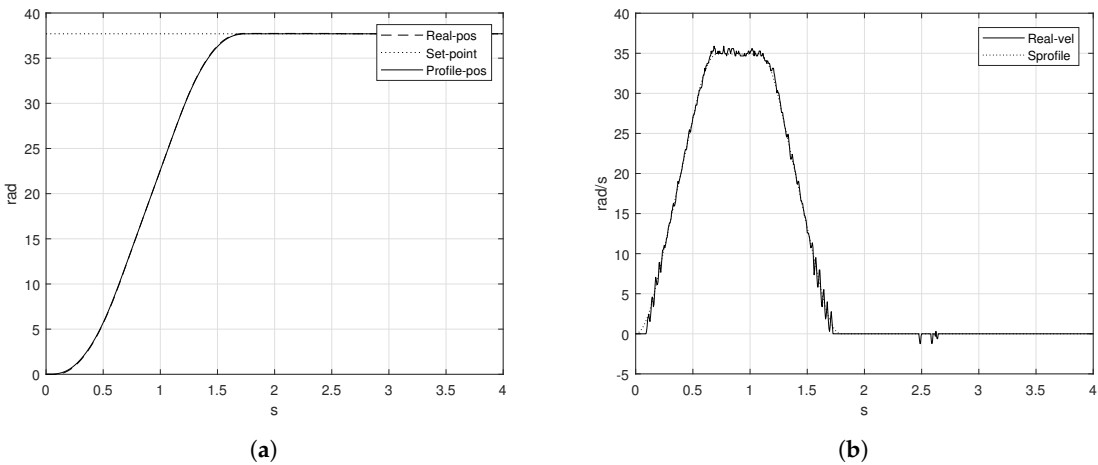

(a)                                                                    (b)

**Figure 11.** S-curve velocity profile implementation (**a**) position and (**b**) velocity.

The speed measured by the rotary encoder follows properly the speed computed by the algorithm. When the velocity reached the constant phase, the seven-segment velocity profile presented a smooth change in the velocity. Besides, the velocity showed in Figure 9b was lower in magnitude than the obtained by the S-curve profile. The value of $\theta_d$ was reached in the proposed time $T$. Figure 11a shows the behavior of the real position of the shaft of the motor, the real position is pretty similar to the computed position.

The error signal presented in Figure 12a exhibited an error when the position must be reached the set-point of $e_s = 0.04$ rad. The control signal is presented in Figure 12b, where the maximum voltage required for all the displacement is $u(t) = 1.5$ V.

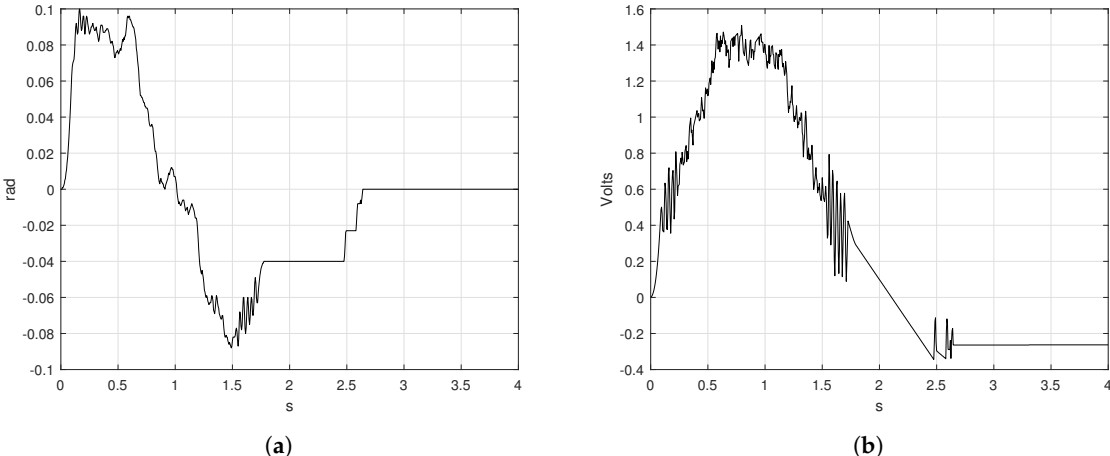

**Figure 12.** Error position and control signals obtained from the S-curve velocity profile implementation (**a**) error and (**b**) control signal.

The acceleration behavior is shown in Figure 13a. The acceleration obtained from the trapezoidal velocity profile shows an aggressive shape due to the acceleration profile is ideally an square signal. Experimentally, the acceleration value was $\alpha_T = \pm 52.3$ rad/s$^2$. The shape of the velocity profile was important to compensate the behavior of the acceleration and avoid discontinuities because of the jerk.

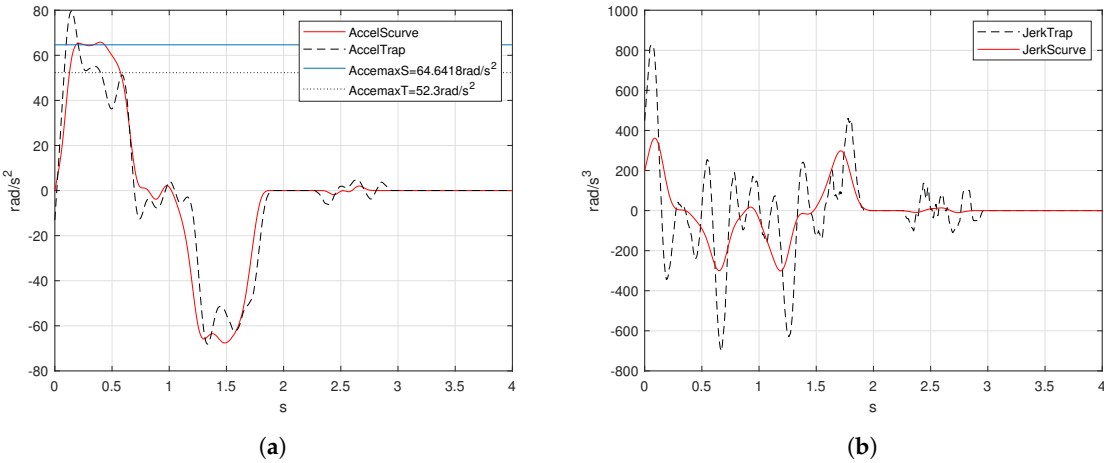

**Figure 13.** Acceleration and jerk of trapezoidal and S-curve motion profiles (**a**) acceleration and (**b**) jerk.

The S-curve velocity profile displays an smooth response in speed and acceleration. The acceleration value computed in (19) is $\alpha_d = 64.6418$ rad/s$^2$. The form of the acceleration computed by the S-curve velocity profile is smoother than the obtained by the trapezoidal. Besides, it maintains the acceleration value computed mathematically.

The mathematical value for the jerk derived from the trapezoidal velocity profile tended to infinity, but for experimental results, the rate of change in acceleration produced an increase in vibrations and the magnitude is relatively high for the acceleration and deceleration phases. For the S-curve velocity profile, the jerk profile is bounded by the designer in (20), so the computed jerk for the experiment was $J_d = 359.1212$ rad/s$^3$. Figure 13b presents the jerk behavior for both velocity profiles. One can prove that the response presented for the S-curve velocity profile was lower in magnitude than the trapezoidal velocity profile. This happens because of the third-degree polynomial proposed for the position profile.

Two positions desired were proposed to prove the negative velocity and displacement constraints in implementation. The values used in the algorithm are presented in Table 3.

**Table 3.** S-curve and inverse S-curve parameters used for implementation.

| Parameters | | Values |
|---|---|---|
| First Displacement | | |
| Desired position | $\theta_{d_1}$ | $12\pi$ rad |
| Time of displacement | $T_1$ | 1.8 s |
| Velocity | $\omega_{d_1}$ | 34.9065 rad/s |
| Acceleration | $\alpha_{d_1}$ | 64.6418 rad/s$^2$ |
| Jerk | $J_{d_1}$ | 359.1212 rad/s$^3$ |
| Second Displacement | | |
| Desired position | $\theta_{d_2}$ | 0 rad |
| Time of displacement | $T_2$ | 1.8 s |
| Velocity | $\omega_{d_2}$ | $-34.9065$ rad/s |
| Acceleration | $\alpha_{d_2}$ | $-64.6418$ rad/s$^2$ |
| Jerk | $J_{d_2}$ | $-359.1212$ rad/s$^3$ |

When the actuator has reached the first set-point, $\theta_{d_1} = 12\pi$, the shaft of the motor was maintained in that point until the new desired position was added. So that, $\theta_0 = \theta_{d_1}$, it means $\theta_0$ is the last point measured by the encoder after $T_1 = 1.8$ s and was the initial position for the next displacement, $T_1$ is the total duration of the first movement. The new position to reach was $\theta_{d_2} = 0$ rad, the aim was to set the shaft of the motor till the origin using the same period of time $T_2 = 1.8$ s. The magnitude of the velocity was the same for both trajectories but the direction was different, so $\omega_{d_2} = -34.9065$ rad/s. The S-curve and the inverse S-curve is presented in Figure 14b. The position profile for both displacements has the shape of the S-curve profile, the position of Figure 14a is displayed by two different movements. The set-points are reached properly in the proposed time. The position measured by the encoder follows the computed position by the algorithm.

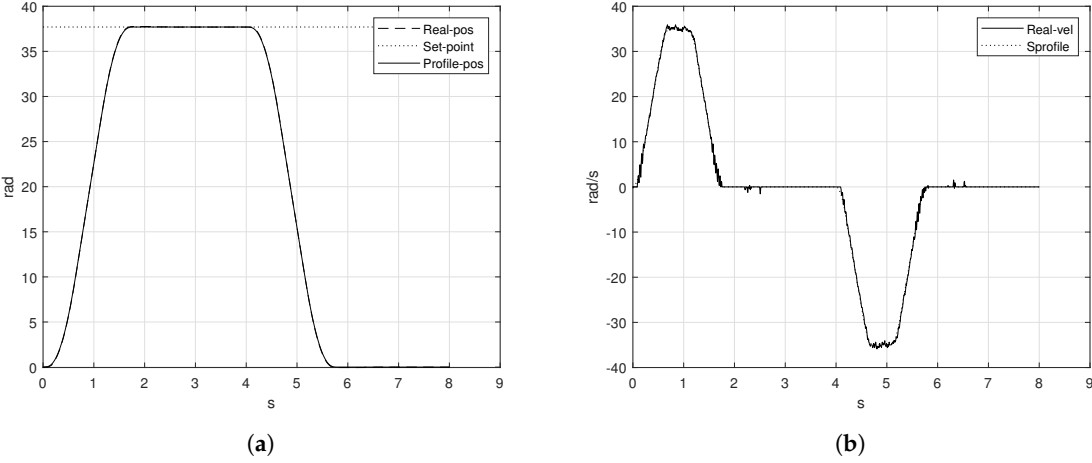

(**a**)             (**b**)

**Figure 14.** S-curve and inverse S-curve velocity profile implementation, (**a**) position, and (**b**) velocity.

The error signal is displayed in Figure 15, the first movement has a duration of $T_1 = 1.8$ s. When $t = T_1$, the error was about $e_s = 0.05$ rad, it means the position is near $\theta_{d_1}$, after 0.47 s the actual error has been decreased to $e_s = 0.009$ rad.

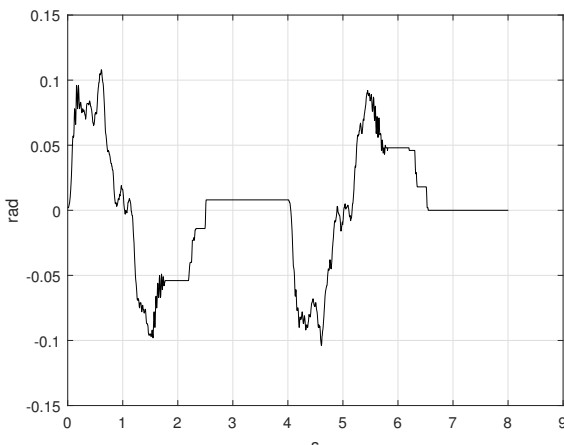

**Figure 15.** Error position signal of the S-curve and inverse S-curve implementation.

The new displacement started in $t = 4$ s. The length of the second movement was $T_2 = 1.8$ s. When $t = 5.8$ s, the present error had a magnitude of $e_{is} = 0.049$ rad, the error turned to zero ($e = 0$ rad) in $t = 6.5$ s according to the Figure 15.

## 5. Conclusions

In this paper, an S-curve velocity profile has been presented for motion control applications implemented in a low-cost platform compounded by a Raspberry Pi 3 and an FPGA. The velocity profile algorithm can be modified to generate a specific trajectory varying the parameters of implementation, such as the length of the acceleration and deceleration phases. Besides, given a desired position $\theta_d$ and the total duration of the movement $T$, one can obtain the magnitude of the speed $\omega_d$, acceleration $\alpha_d$ and the jerk $J_d$ to generate the S-curve velocity profile. The error obtained with the trapezoidal velocity profile was $e_T = 0.18$ rad when the duration of the movement is $T = 1.8$ s. The error obtained with the application of the S-curve velocity profile is about $e_s = 0.05$ rad in the same time.

Notice that the magnitude of the speed used in the S-curve was $\omega_d = 34.9065$ rad/s and $\omega_{d_T} = 32$ rad/s for the trapezoidal velocity profile. Despite the maximum speed reached for the trapezoidal profile is less in magnitude than the S-curve profile, the velocity behavior does not follow properly the computed speed by the algorithm because of the rate of change in acceleration. The S-curve velocity profile presents a smoother change in the position than the trapezoidal profile due to the third-degree polynomial proposed in the acceleration-deceleration stages.

**Author Contributions:** Conceptualization, J.R.G.-M. and J.R.-R.; methodology, J.R.G.-M.; software, J.R.G.-M.; validation, J.R.G.-M., E.E.C.-M. and J.R.-R.; formal analysis, J.R.G.-M.; investigation and visualization, J.R.G.-M., E.E.C.-M. and J.R.-R.; data curation, J.R.G.-M., E.E.C.-M. and J.R.-R.; writing—original draft preparation; writing—original draft, review, and editing, all the authors.

**Funding:** This research was funded by the "Consejo Nacional de Ciencia y Tecnología (CONACYT)" under the scholarship 778619.

**Acknowledgments:** We would like to thank the Graduate Studies Division from the Faculty of Engineering at Universidad Autónoma de Querétaro by allowing me to make Ph.D. studies.

**Conflicts of Interest:** The authors declare no conflict of interest.

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
