# Peer review of "A New Seven-Segment Profile Algorithm for an Open Source Architecture in a Hybrid Electronic Platform"

_electronics, doi:10.3390/electronics8060652_

Round 1

Reviewer 1 Report

CONTRIBUTION: 

The key contribution presented in this manuscript is a novel methodology to obtain the coefficients of the S-curve for the velocity profile for point-to-point motions in real-time. The path, motion duration, and jerk can be controlled with the S-curve coefficients. The motivation for this is to have smoother trajectories and thereby reduce the wear of the robot drive-train and structure. 

GENERAL COMMENTS: 

I. Please clarify what the specific goal, objectives and contributions are for the work presented in this manuscript in the introduction. 

II. Please highlight clearly what the novelty is of the presented work in this manuscript. 

III. As is also discussed in the manuscript, there has been a significant amount of previous research been conducted on this topic. Hence, it is necessary that the methodology proposed in this manuscript, is compared against the most advanced (recent) state-of-the-art methodologies. At the moment, the trajectory generated according to the proposed methodology is only compared against a trapezoidal trajectory but there is no comparison against other existing methodologies to generate S-curve trajectories. 

Please also look into other relevant work such as the ones listed below in order to compare your proposed approach against other methodologies to optimise trajectories in order to minimise the wear of robots. This will help to clearly identify the main contributions with the work presented in this manuscript. For example: 

- A. Gasparetto, V. Zanotto, A technique for time-jerk optimal planning of robot trajectories, Robotics and Computer-Integrated-Manufacturing, Vol.24 (3), pp. 415-426, 2008

- E. Glorieux, B. Svensson, F. Danielsson, B. Lennartson, Multi-objective constructive cooperative coevolutionary optimization of robotic press-line tending, Engineering Optimization, Vol. 49 (10), pp. 1685-1703, 2017

IV. A thorough revision of the English language is very much necessary to improve the manuscript in order to make it suitable for publication. 

SPECIFIC COMMENTS: 

INTRODUCTION: 

1. Last sentence in first paragraph: Please revise the statement that a high jerk indicates a high energy consumption, since the kinetic energy is determined by the torque and the velocity. 

2. First sentence of the second paragraph: Please perform a broader literature review regarding different velocity profiles (interpolating polynomials, parabolic blends, bezier curves, etc.). I would suggest you also consult textbooks such as B. Siciliano, L. Sciavicco, L. Villani, G. Oriolo, Robotics modelling, planning and control, Springer, 2009

3. Please explain the abbreviation FPGA, the first time it is used in the text. 

4. What is the significance of being able to obtain S-curve coefficients in real-time in comparison with other existing methodologies to generate S-curves? 

BACKGROUND:

5. Second to last sentence in first paragraph: Please clarify in the text why you mean with "when the degree of the polynomial is increased, the profiles tend to shift themselves in a particular position". 

Section 3: 

6. Please explain the abbreviations or codes XILINX, ZYNQ XC7Z010 in the text to the readers as well as why these architectures are being used in this work (i.e. from which technology advances in industry that have arisen). 

7. Please explain the depicted architecture in Figure 4 in the text in order to present it to the readers.

8. Please explain the entities depicted in Figure 5 in the text in order to present the information/message to the reader. 

9. Please explain why "for this application the total time of the acceleration is devided by \gamma = 0.4, and T_{acc}=\gamma T" in the text.

SECTION 4

10. Please provide a reference for the statement that "The trapezoidal velocity profile is the most used in industrial applications due to the ease of implementation since it consists of two linear equations describing the acceleration-deceleration phases, and a constant velocity stage." because I have some doubts that any (commercial) robots use this type of velocity profile. Maybe it is better to rephrase this statement instead. Please thoroughly investigate the velocity profiles used for real-world robot applications in industry. 

11. Please clarify how the tests have been conducted and for what purpose. In other words, please state in the text for which reasons the specific tests have been conducted and motive the design of the tests/experiments accordingly.

Author Response

Authors appreciate the corrections and hope that the answers enlisted below in this document fulfill the expectations. We considered every suggestion as a valuable opportunity to improve and enrich enormously our work. Therefore, we have intended to be as punctual as we could to attend the decision of Minor Revisions. Reviewer observations are highlighted using italic red font, our replies using blue font.

GENERAL COMMENTS:

I. Please clarify what the specific goal, objectives and contributions are for the work presented in this manuscript in the introduction.

We appreciate the comment. We are mentioning in the abstract: “Furthermore, a new methodology is developed to generate a seven-segment profile that works with negative velocity and displacement constraints applying an open source architecture in a hybrid electronic platform compounded by a System on a Chip (SoC) Raspberry Pi 3 and a Field Programmable Gate Array (FPGA).” Also, in the Introduction, we say, “According to a comparison presented in [2], the controller band the profile generator algorithm can be chosen by the designer according to its experience…”.

II. Please highlight clearly what the novelty is of the presented work in this manuscript.

Thank you for the comment. We are mentioning in the abstract: “Furthermore, a new methodology is developed to generate a seven-segment profile that works with negative velocity and displacement constraints applying an open source architecture in a hybrid electronic platform compounded by a System on a Chip (SoC) Raspberry Pi 3 and a Field Programmable Gate Array (FPGA).”

III. As is also discussed in the manuscript, there has been a significant amount of previous research been conducted on this topic. Hence, it is necessary that the methodology proposed in this manuscript, is compared against the most advanced (recent) state-of-the-art methodologies. At the moment, the trajectory generated according to the proposed methodology is only compared against a trapezoidal trajectory but there is no comparison against other existing methodologies to generate S-curve trajectories.

We appreciate the comment. The algorithm proposed in this paper has been compared with the results presented in [31], the input data has been displayed in Table 2.

Please also look into other relevant work such as the ones listed below in order to compare your proposed approach against other methodologies to optimise trajectories in order to minimise the wear of robots. This will help to clearly identify the main contributions with the work presented in this manuscript. For example:

- A. Gasparetto, V. Zanotto, A technique for time-jerk optimal planning of robot trajectories, Robotics and Computer-Integrated-Manufacturing, Vol.24 (3), pp. 415-426, 2008

- E. Glorieux, B. Svensson, F. Danielsson, B. Lennartson, Multi-objective constructive cooperative coevolutionary optimization of robotic press-line tending, Engineering Optimization, Vol. 49 (10), pp. 1685-1703, 2017

We appreciate the comment. We took into account contemporary papers since this topic is growing fast. Then, we compare our proposal with trapezoidal and s-curve methods. In that way, we add certain references regarding this approach.

IV. A thorough revision of the English language is very much necessary to improve the manuscript in order to make it suitable for publication.

Thank you for the comment. We sent our paper to revision with a native language speaker.

SPECIFIC COMMENTS:

INTRODUCTION:

1. Last sentence in first paragraph: Please revise the statement that a high jerk indicates a high energy consumption, since the kinetic energy is determined by the torque and the velocity.

Thank you for the comment. You are right, the kinetic energy is determined by the torque and the velocity. However, there is a link between torque, velocity, and jerk; that is the reason why we wrote that sentence.

2. First sentence of the second paragraph: Please perform a broader literature review regarding different velocity profiles (interpolating polynomials, parabolic blends, bezier curves, etc.). I would suggest you also consult textbooks such as B. Siciliano, L. Sciavicco, L. Villani, G. Oriolo, Robotics modelling, planning and control, Springer, 2009

We appreciate the comment. In the new version of our manuscript, those references have been included.

3. Please explain the abbreviation FPGA, the first time it is used in the text.

We appreciate the comment. In the new version of our manuscript, it is defined in the Abstract.

4. What is the significance of being able to obtain S-curve coefficients in real-time in comparison with other existing methodologies to generate S-curves?

Thank you for the comment. In our proposal, S-curve coefficients are calculated in real time which implies that the system is going to have better performance than the simply S-curves. That comparison is carried out in Table 2.

BACKGROUND:

5. Second to last sentence in first paragraph: Please clarify in the text why you mean with "when the degree of the polynomial is increased, the profiles tend to shift themselves in a particular position".

Thank you for the comment. We rewrite the sentence in order to clarify the idea.

Section 3:

6. Please explain the abbreviations or codes XILINX, ZYNQ XC7Z010 in the text to the readers as well as why these architectures are being used in this work (i.e. from which technology advances in industry that have arisen).

Thank you for the comment. In the current version, we rewrite: “Figure 4 to design a motion controller using a hybrid system compounded by the interaction of a Raspberry Pi 3 and an FPGA Zybo-ZYNQ XC7Z010 from the family XILINX…”.

7. Please explain the depicted architecture in Figure 4 in the text in order to present it to the readers.

We appreciate the comment. In the current version, we rewrite: “The processing and control systems are embedded on the Raspberry Pi 3. The seven-segment velocity profile is developed in C programming;…”.

8. Please explain the entities depicted in Figure 5 in the text in order to present the information/message to the reader.

We appreciate the comment. In the current version we add: “On the other hand, the FPGA reads the real position of the shaft using a rotary encoder (2000 Pulses Per Revolution)…”.

9. Please explain why "for this application the total time of the acceleration is devided by \gamma = 0.4, and T_{acc}=\gamma T" in the text.

We appreciate the comment. We update the text to say: “for this application the total time of the acceleration is divided by \gamma = 0.4, and T_{acc}=\gamma T In order to have a simetric profile”.

SECTION 4

10. Please provide a reference for the statement that "The trapezoidal velocity profile is the most used in industrial applications due to the ease of implementation since it consists of two linear equations describing the acceleration-deceleration phases, and a constant velocity stage." because I have some doubts that any (commercial) robots use this type of velocity profile. Maybe it is better to rephrase this statement instead. Please thoroughly investigate the velocity profiles used for real-world robot applications in industry.

We appreciate the comment. We added “Yoon, H.; Chung, S.; Kang, H.; Hwang, M. Trapezoidal Motion Profile to Suppress Residual Vibration of Flexible Object Moved by Robot. Electronics 2019, 8, 30.”

11. Please clarify how the tests have been conducted and for what purpose. In other words, please state in the text for which reasons the specific tests have been conducted and motive the design of the tests/experiments accordingly.

We appreciate the comment. We made a test bench to validate the low cost, high-efficiency of the motion control system.

Reviewer 2 Report

The paper presents a method to obtain the S-curve coefficients in real time that could integrates actuator speed and displacement constraints and such method has been applied to a DC motor on Raspberry pi 3 and FPGA implementation.

The paper is well organized and clear. The results seem very interest for the aim and scope of the Electronics.  In my opinion, the paper is accepted in present form after minor revision.

The paper said that S-curve velocity curve is generally better than trapezoidal in the introduction section, then the author need not to spend too much efforts to compare these two as results are well-know. Instead author could provide more analysis to compare the proposed method to obtain S-curve coefficients as with common method to get these.

Author Response

Authors appreciate the corrections and hope that the answers enlisted below in this document fulfill the expectations. We considered every suggestion as a valuable opportunity to improve and enrich enormously our work. Therefore, we have intended to be as punctual as we could to attend the decision of Minor Revisions. Reviewer observations are highlighted using italic red font, our replies using blue font.

The paper presents a method to obtain the S-curve coefficients in real time that could integrates actuator speed and displacement constraints and such method has been applied to a DC motor on Raspberry pi 3 and FPGA implementation.

The paper is well organized and clear. The results seem very interest for the aim and scope of the Electronics.  In my opinion, the paper is accepted in present form after minor revision.

The paper said that S-curve velocity curve is generally better than trapezoidal in the introduction section, then the author need not to spend too much efforts to compare these two as results are well-know. Instead author could provide more analysis to compare the proposed method to obtain S-curve coefficients as with common method to get these.

We appreciate the comment. The algorithm proposed in this paper has been compared with the results presented in [31], the input data has been displayed in Table 2.

Reviewer 3 Report

The paper is well written. The experimental part in section 4 may be further elaborated to show the implementation results. 

Author Response

Authors appreciate the corrections and hope that the answers enlisted below in this document fulfill the expectations. We considered every suggestion as a valuable opportunity to improve and enrich enormously our work. Therefore, we have intended to be as punctual as we could to attend the decision of Minor Revisions. Reviewer observations are highlighted using italic red font, our replies using blue font.

The paper is well written. The experimental part in section 4 may be further elaborated to show the implementation results.

We appreciate the comment. In the current version Figures 9-12 shows the experimental result, and moreover, the algorithm proposed in this paper has been compared with the results presented in [31], the input data has been displayed in Table 2.

Round 2

Reviewer 2 Report

I appreciated the modifications and the added comparation with the results presented in [33] in the section 4 Simulation and Results. However, I suggest a more detailed comparation and discussion such as the improvements and indications.  

In page 11, “ Unlike the jerk values obtained with the Yi Fang and Wenhai Liu algorithm Jmax = 20,…” I see that the Jerk value with the Yi Fang algorithm should be 30 instead of 20 from Table 2 based on same initial and final points. Also, don’t quite understand how is proposed method better than the quoted method with Yi Fang.

Author Response

Dr. Michelle Zhou

Managing Editor MDPI Electronics

Prof. Dr. Cecilio Angulo

Guess Editor MDPI Electronics

We are pleased to resend you the article A New Seven-Segment Profile Algorithm for an Open Source Architecture in a Hybrid Electronic Platform, for your consideration, with the good intention to be published in the prestigious journal: MDPI Electronics- Cognitive Robotics & Control.

Authors appreciate the corrections and hope that the answers enlisted below in this document fulfill the expectations. We considered every suggestion as a valuable opportunity to improve and enrich enormously our work. Therefore, we have intended to be as punctual as we could to attend the decision of Minor Revisions. Reviewer observations are highlighted using italic red font, our replies using blue font.

The purpose of the project was to implement a new methodology to generate a seven-segment profile that works with negative velocity and displacement constraints applying an open source architecture in a hybrid electronic platform compounded by a System on a Chip (SoC) Raspberry Pi 3 and a Field Programmable Gate Array (FPGA). This is a very useful algorithm that can be used in an industrial process to avoid high stress in the motor. Thus, it is necessary to design smoother movements. It is shown the rate of change in acceleration with the S-curve velocity profile, which presents an acceleration and deceleration stage smoother than the trapezoidal velocity profile reducing the error at the end of the duty-cycle pre-established. The performance of the motion controller is measured through the comparison of the error obtained in a real-time application with a trapezoidal velocity profile. As a result, a low-cost platform and an open architecture system are achieved.

We would also like to remark that this is an unpublished paper, submitted only to your prestigious Journal, and the manuscript has not been published elsewhere in any form, nor as proceeding or any other divulgation media.

Dr. Juvenal Rodríguez Reséndiz,

IEEE Senior Member

IEEE Queretaro Section Past President,

(corresponding autor)

juvenal@ieee.org,

www.juvenal.mx,

Universidad Autónoma de Querétaro, México,

Cerro de las campanas S/N,

Campanas, Z. 76010,

Querétaro Qro.  Tel.  +52 (442) 192 12 00 ext. 3591

Reviewer 2

I appreciated the modifications and the added comparation with the results presented in [33] in the section 4 Simulation and Results. However, I suggest a more detailed comparation and discussion such as the improvements and indications.

We appreciate the comment. We added to the paper “the acceleration stage in [33] is compounded by seven intervals, while the proposed method in this paper has three phases showing better simulation results”.

In page 11, “ Unlike the jerk values obtained with the Yi Fang and Wenhai Liu algorithm Jmax = 20,…” I see that the Jerk value with the Yi Fang algorithm should be 30 instead of 20 from Table 2 based on same initial and final points. Also, don’t quite understand how is proposed method better than the quoted method with Yi Fang.

Thank you for the comment. Yi Fang method proposed a fourth order polynomial, while the present paper uses a third order polynomial. However, the performance of our proposal is better of the Yi Fang as shows Table 2.
